# Shuffle the Context: RoPE-Perturbed Self-Distillation for Long-Context Adaptation

**Zichong Li** [* 1]  **Chen Liang** [2]  **Liliang Ren** [2]  **Tuo Zhao** [1]  **Yelong Shen** [2]  **Weizhu Chen** [2]

## Abstract

Large language models (LLMs) increasingly operate in settings that require reliable long-context understanding, such as retrieval-augmented generation and multi-document reasoning. A common strategy is to fine-tune pretrained short-context models at the target sequence length. However, we find that standard long-context adaptation can remain brittle: model accuracy depends strongly on the absolute placement of relevant evidence, exhibiting high positional variance even when controlling for task format and difficulty. We propose *RoPE-Perturbed Self-Distillation*, a training regularizer that improves positional robustness. The core idea is to form alternative "views" of the same training sequence by perturbing its RoPE indices—effectively moving parts of the context to different positions—and to train the model to produce consistent predictions across views via self-distillation. This encourages reliance on semantic signals instead of brittle position dependencies. Experiments on long-context adaptation of Llama-3-8B and Qwen-3-4B demonstrate consistent gains on long-context benchmarks, including up to 12.04 percent-point improvement on RULER-64K for Llama-3-8B and 2.71 percent-point gain on RULER-256K for Qwen-3-4B after SFT, alongside improved length extrapolation beyond the training context window.

## 1. Introduction

Many real-world applications of large language models (LLMs)—including long-horizon software engineering and

reasoning (Jimenez et al., 2024), multi-document question answering (Bai et al., 2024), and retrieval-augmented generation (RAG) (Gao et al., 2023)—critically depend on strong long-context performance. In these settings, relevant evidence may appear anywhere within very long input sequences and may shift substantially across queries due to document ordering, concatenation strategy, and retrieval placement.

A common way to improve long-context performance is to fine-tune short-context pretrained LLMs on sequences at the target length (Yang et al., 2025a; Team et al., 2024), often monitoring progress using synthetic needle-in-a-haystack (NIAH) evaluations (Hsieh et al., 2024; Dubey et al., 2024; Xiaomi, 2026). Despite the effectiveness of such long-context fine-tuning (Gao et al., 2025), we find that the resulting models can remain brittle with respect to evidence placement. Figure 1a illustrates this effect on the RULER multikey-2 NIAH task (Hsieh et al., 2024): a strong long-context fine-tuned baseline exhibits a pronounced position dependence, with accuracy varying with the answer's location within the prompt. While this behavior does not preclude good average performance, it is undesirable in long-context pipelines such as RAG, codebase/file concatenation, or multi-source packing, where the same evidence span may appear at widely different absolute positions across inputs.

This position sensitivity can be viewed through the lens of rotary position embeddings (RoPE; (Su et al., 2024)), which most modern LLMs use for positional encoding. RoPE rotates query and key vectors by position-dependent angles, so attention depends on relative offsets. In long-context regimes, these position-dependent phases can make model behavior sensitive to how positional indices are assigned.

This perspective motivates explicitly encouraging *positional robustness* during training. We propose *RoPE-Perturbed Self-Distillation*, a simple regularizer that creates an alternative 'view" of each training sequence by shifting RoPE indices for a contiguous span (Figure 1b). Concretely, given a length-$L$ sequence, we sample a split point $s$ and add an offset $y$ to the RoPE indices of all tokens in the suffix $[s, L-1]$, while keeping $[0, s-1]$ unchanged. This perturbation effectively increases the RoPE distance between the suffix and the earlier prefix by $y$, so tokens in $[s, L-1]$ 'see"

---

[*]Work is done during internship at Microsoft. [1]Georgia Institute of Technology [2]Microsoft. Correspondence to: Zichong Li <zli911@gatech.edu>, Chen Liang <chen-liang1@microsoft.com>.

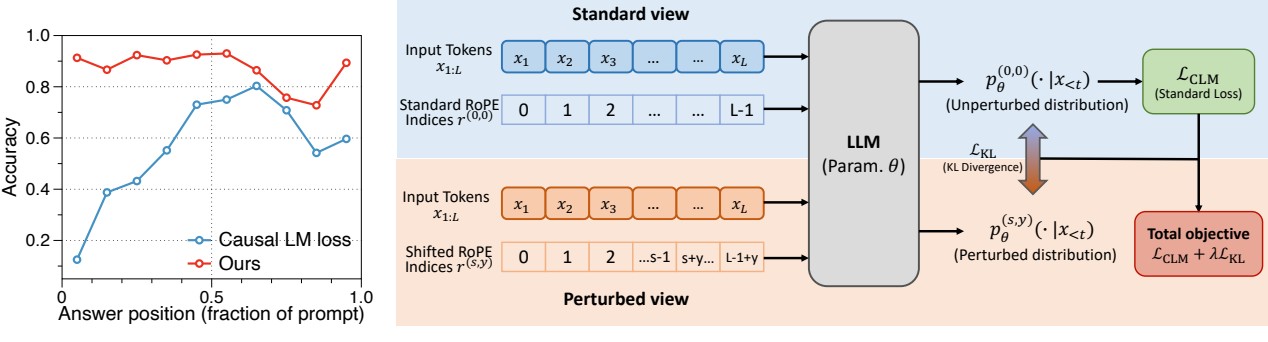

*(a)* Answer position vs. accuracy.                                        *(b)* Overview.

*Figure 1.* (a) Relation between answer position and accuracy on the NIAH multikey-2 task from RULER (Hsieh et al., 2024) with 64k-token sequences. We evaluate the ProLong-64k-base model (Gao et al., 2025) as causal LM baseline and model trained with our method. (b) Overview of our proposed objective.

the preceding context as farther away.

We train the model with two forward passes: a *standard view* and a *perturbed view* (Figure 1b). On the standard view, we optimize the usual causal language modeling (CLM) loss. We then enforce cross-view consistency by minimizing a Kullback–Leibler (KL) divergence that matches the perturbed-view next-token distribution to the standard-view distribution, using stop-gradient on the standard path. In this way, the standard view provides a stable reference under the original index assignment, while the perturbed view is trained to align with it, penalizing prediction differences attributable purely to the RoPE shift and thereby reducing sensitivity to evidence placement.

The effect of this perturbation can be understood through RoPE's multi-frequency structure: higher-frequency components vary rapidly with position (capturing fine-grained phase), while lower-frequency components vary more slowly (capturing coarser structure). The index shift primarily disrupts the position-sensitive high-frequency components for interactions involving the shifted span, while leaving coarse structure relatively stable. Enforcing agreement across the two views therefore encourages the model to rely more on position-robust signals—semantic content and coarse positional structure.

Empirically, as shown in Figure 1a, our RoPE-perturbed self-distillation substantially reduces the variation of accuracy with answer position on the NIAH task, suggesting reduced positional brittleness. Complementing this, our attention analysis on long sequences (Figure 2) shows that models trained with our objective allocate more attention mass to larger distances and exhibit a more uniform distribution of long-range attention.

In Section 3, we present a more comprehensive evaluation on long-context adaptation of Llama-3-8B and Qwen-3-4B using the RULER (Hsieh et al., 2024) and HELMET

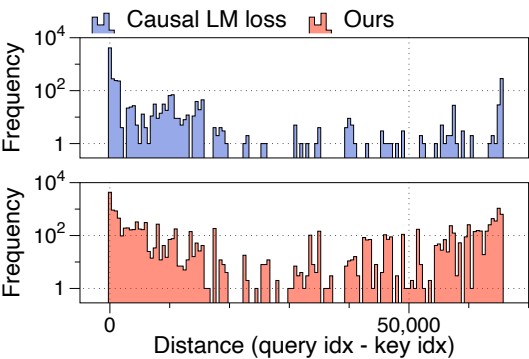

*Figure 2.* Analysis of attention distance patterns on a 64k-token sequence from Prolong (Gao et al., 2025) dataset. For both the baseline model and our model, we inspect attention scores from the final 256 query tokens. We consider attention weights above a threshold ($10^{-3}$) and measure the signed distance between query and key indices. The figure shows the histogram of attention distances in the 24th layer.

(Yen et al., 2025) benchmarks. Across both model families, our method yields consistent gains on long-context benchmarks, including 12.04% on RULER-64K for Llama-3-8B and 2.71% on RULER-256K for Qwen-3-4B after SFT, with similarly positive trends on HELMET. We further observe improved length extrapolation when evaluating at 2–4× the training context length using YaRN-based extension (Peng et al., 2024), supporting robustness beyond the trained window. Finally, these improvements come with comparable short-context performance to the baseline.

## 2. Method

### 2.1. Problem setting and notation

Let $x_{0:L-1} = (x_0, \ldots, x_{L-1})$ be a token sequence of length $L$ drawn from a long-context corpus $\mathcal{D}$. We consider an autoregressive Transformer language model with parameters

$\theta$ that uses rotary position embeddings (RoPE; (Su et al., 2024)) in each self-attention layer. We expose RoPE through an explicit index vector $r = (r_0, \ldots, r_{L-1}) \in \mathbb{Z}^L$, where token $x_i$ is assigned RoPE index $r_i$. Given $(x_{0:L-1}, r)$, the model defines next-token distributions

$$p_\theta(\cdot \mid x_{<i}; r), \qquad i = 0, \ldots, L-1, \qquad (1)$$

under a causal mask (so token $i$ attends only to $< i$). Standard long-context fine-tuning minimizes the causal language modeling (CLM) loss under the unperturbed RoPE indices:

$$r_i = i, \qquad i = 0, \ldots, L-1, \qquad (2)$$

$$\mathcal{L}_{\text{CLM}}(\theta) = \mathbb{E}_{x_{0:L-1} \sim \mathcal{D}} \left[ -\sum_{i=0}^{L-1} \log p_\theta(x_i \mid x_{<i}; r) \right]. \quad (3)$$

Our goal is to adapt LLMs to long contexts while reducing brittle dependence on RoPE indices. We do so by constructing *perturbed views* of the same token sequence—views that keep content and the causal mask unchanged but alter RoPE indices—and enforcing prediction consistency across views via self-distillation. The framework is essentially agnostic to the specific perturbation: any transformation that changes RoPE indices can serve as a view generator.

### 2.2. RoPE-perturbed views via skip-based index shifts

**Skip-based perturbation.** We define a family of RoPE index assignments parameterized by a split point $s$ and a skip length $y$:

$$r_i^{(s,y)} = \begin{cases} i, & i < s, \\ i + y, & i \ge s, \end{cases} \qquad i = 0, \ldots, L-1, \qquad (4)$$

where $s \in \{0, \ldots, L-1\}$ and $y \in \{1, \ldots, Y\}$. For each sequence, we sample $(s, y) \sim q(s, y)$ and apply the same indices across all RoPE layers. We set $Y = L$ and $q$ to be uniform distribution in our main experiments. We explore the effect of changing $Y$ in section 3.6.

Crucially, the token sequence $x_{0:L-1}$ is unchanged; only RoPE indices are modified. Intuitively, this creates a "gap" in index space: tokens in the suffix $[s, L-1]$ are assigned larger absolute RoPE indices, making cross-boundary interactions between the prefix $[0, s-1]$ and suffix $[s, L-1]$ appear farther apart in RoPE space.

**Two-view construction.** For each input $x_{0:L-1}$ and sampled $(s, y)$, we form:

- a *standard view*: $p_\theta^{(0,0)}(\cdot \mid x_{<i}) = p_\theta(\cdot \mid x_{<i}; r^{(0,0)})$,

- a *perturbed view*: $p_\theta^{(s,y)}(\cdot \mid x_{<i}) = p_\theta(\cdot \mid x_{<i}; r^{(s,y)})$,

for all $i = 0, \ldots, L-1$. We write the standard RoPE indices as a special case $r^{(0,0)}$. Here, standard view and perturbed view refer to two forward passes of the same model on the same token sequence, differing only in the positional index assignment.

### 2.3. Self-distillation between perturbed and standard views

We encourage *positional robustness* by enforcing that the model's predictions under perturbed RoPE indices match those under the standard indices, while keeping the standard forward pass as the reference. Concretely, we minimize a *reverse* KL divergence, which we find empirically performs better than the forward KL. For a target position $i$, we define

$$\ell_i^{(s,y)}(\theta) = \text{KL}\left( p_\theta^{(s,y)}(\cdot \mid x_{<i}) \,\middle\|\, \text{sg}\left( p_\theta^{(0,0)}(\cdot \mid x_{<i}) \right) \right), \quad (5)$$

where $\text{sg}(\cdot)$ denotes stop-gradient so gradients from the regularizer flow only through the perturbed path. Intuitively, the standard view acts as a teacher: it is directly optimized by the CLM objective under the original indexing and thus provides a stable target distribution, while the perturbed view is trained to match this reference under shifted indices.

Because the perturbation in Eq. (4) leaves all RoPE indices unchanged for positions $< s$, the two views are identical for prefix targets, implying $\ell_i^{(s,y)}(\theta) = 0$ for all $i < s$. We therefore apply the distillation loss only on suffix targets:

$$\mathcal{L}_{\text{distill}}(x_{0:L-1}; \theta) = \mathbb{E}_{(s,y) \sim q(s,y)} \left[ \frac{1}{L-s} \sum_{i=s}^{L-1} \ell_i^{(s,y)}(\theta) \right]. \quad (6)$$

Finally, we take expectation over the data distribution:

$$\mathcal{L}_{\text{KL}}(\theta) = \mathbb{E}_{x_{0:L-1} \sim \mathcal{D}} \left[ \mathcal{L}_{\text{distill}}(x_{0:L-1}; \theta) \right]. \quad (7)$$

**Overall objective.** Our full training objective combines standard CLM fine-tuning with the RoPE-perturbed self-distillation regularizer:

$$\mathcal{L}_{\text{total}}(\theta) = \mathcal{L}_{\text{CLM}}(\theta) + \lambda \mathcal{L}_{\text{KL}}(\theta), \qquad (8)$$

and we use $\lambda = 1$ in all experiments for simplicity. Operationally, the standard view is updated via $\mathcal{L}_{\text{CLM}}$, improving the teacher distribution over the course of training, while the perturbed view is updated via the KL regularizer to match this evolving teacher and become invariant to the controlled index shift. Relative to standard long-context fine-tuning, our method adds one additional forward pass per batch (the perturbed view). We analyze the wall-clock overhead and compare performance under matched compute in Section 3.7.

At evaluation time we report performance under the standard (unperturbed) indexing. Thus, the regularizer does not introduce a second model; it simply encourages a single model to produce consistent predictions under controlled positional perturbations.

### 2.4. Cyclic-shift perturbation as a working variant

The skip-based shift in Eq. (4) is one concrete perturbation. More broadly, our objective is to encourage invariance to index changes, and many RoPE-index perturbations can instantiate this principle. In addition to skip-based shifts, we also consider a cyclic-shift perturbation, which we find empirically effective as well.

Given a shift $u \in \{0,\ldots,L-1\}$, define:

$$r_i^{\text{cyc}(u)} = (i + u) \bmod L, \quad i = 0,\ldots,L-1. \tag{9}$$

We then define the perturbed view as $p_\theta^{\text{cyc}(u)}(\cdot \mid x_{<i}) = p_\theta(\cdot \mid x_{<i}; r^{\text{cyc}(u)})$ and apply the same distillation loss in Eq. (6) by sampling $u \sim q(u)$.

Unlike the skip-based shift, which preserves the relative order of tokens, a cyclic shift rotates the index assignment so that tokens in the original prefix are mapped to the end of the sequence (and vice versa). As a result, this perturbation is more destructive and can substantially change long-range relative offsets for many token pairs. From the perspective of our objective, cyclic shift therefore serves as a stronger stress test of index invariance.

## 3. Experiments

We evaluate whether RoPE-perturbed self-distillation improves (i) long-context performance at the trained window, (ii) robustness under length extrapolation, and (iii) practicality under compute and post-training (SFT), while preserving short-context capability. More training details and results are deferred to Appendix A and B.

### 3.1. Experimental Setup

**Models.** We study long-context adaptation on two LLMs: **Llama-3-8B-Instruct** (default 8K window) and **Qwen-3-4B** (default 32K window). We extend the maximum context length to 64K for Llama and 256K for Qwen. Following ProLong (Gao et al., 2025), we use the instruction-tuned variants to obtain stronger downstream behavior and more stable long-context evaluation (e.g., better instruction following and reduced formatting failures), which improves the reliability of benchmark measurements. Following prior work, we use ABF RoPE extrapolation (Xiong et al., 2024) and increase the RoPE base to $8 \times 10^6$ (Llama) and $1 \times 10^7$ (Qwen); ABF is used consistently for both training and in-distribution evaluation. Unless otherwise stated, all continued-pretraining and SFT experiments use full-parameter updates.

**Training data.** We apply the training dataset used in Pro-Long (Gao et al., 2025). For Llama, we train on the fixed-length 64K `book` subset of ProLong-64K for 4B tokens.

For Qwen, we train on ProLong-512K `book` and `repo` subsets truncated to 256K for 8B tokens. We match Pro-Long hyperparameters (optimizer, schedule, etc.) and train for 1000 steps in both settings. Unless stated otherwise, our default perturbation is the **skip-based shift** (Section 2.2), sampled per-sample. We also report the cyclic-shift variant in the main results.

**Baselines.** We compare against: (i) **Standard**: Causal-LM (CLM)-only long-context fine-tuning; (ii) **LongCE**: likelihood-ratio-based token reweighting (Fang et al., 2025); (iii) **PoSE** (Zhu et al., 2023): positional skip-wise training, implemented as CLM training on the same perturbed view used by our method.

**Evaluation benchmarks.** We primarily evaluate on RULER (Hsieh et al., 2024) and HELMET (Yen et al., 2025), which are designed to probe long-context robustness and behaviors. **RULER** (Hsieh et al., 2024) is one of the most widely used benchmarks for long-context evaluation, covering a broad suite of tasks spanning needle-in-a-haystack retrieval, multi-value/multi-query variants, question answering, and other context-sensitive settings. We report accuracy for each task and the unweighted average, evaluating at {32K, 64K} for Llama and {128K, 256K} for Qwen. **HELMET** (Yen et al., 2025) includes task families designed to better reflect practical long-context use cases (e.g., RAG and ICL). We report the HELMET categories that use non-model-based metrics. To partially bridge toward more realistic downstream usage, we additionally test whether gains persist after instruction SFT and report results on LongBench-v2 (Bai et al., 2024) in section 3.4. We also evaluate short-context performance on standard downstream benchmarks (MMLU (Hendrycks et al., 2021), HellaSwag (Zellers et al., 2019), WinoGrande (Sakaguchi et al., 2020), OpenBookQA (Mihaylov et al., 2018)) in section 3.5.

### 3.2. Main Results on Long-Context Benchmarks

Tables 1–3 summarize our main results across context lengths for Llama-3-8B-Instruct and Qwen-3-4B.

**Llama-3-8B-Instruct.** On Llama at 64K RULER, our method substantially improves the RULER average relative to Standard and all baselines, and also yields consistent gains at 32K (Table 1). The same trend holds on HELMET (64K): our skip-based objective achieves the best average across the reported categories, indicating that the improvements transfer across tasks.

To understand where the gains come from, Table 2 reports task-level breakdown on RULER at 64K. The largest improvements concentrate on more challenging long-context retrieval and composition settings, particularly multi-key retrieval and aggregation-heavy tasks. This aligns with our motivation: enforcing prediction invariance under RoPE

*Table 1.* Average performance of trained Llama-3-8B-Instruct under different sequence lengths on RULER and HELMET (64K) benchmarks.

| Method | RULER | | HELMET | | | | |
|---|---|---|---|---|---|---|---|
| | 32K | 64K | RAG | ICL | LongQA | Rerank | Avg. |
| Standard | 85.23 | 57.25 | 59.08 | 86.30 | 30.30 | 1.51 | 44.30 |
| LongCE | 86.11 | 52.51 | 54.71 | 85.80 | 31.64 | 11.20 | 45.84 |
| PoSE | 87.20 | 67.47 | 57.22 | 86.30 | 30.23 | 8.85 | 45.65 |
| Ours (cyclic shift) | 87.33 | **71.30** | 58.34 | 86.80 | 30.63 | 8.70 | 46.12 |
| Ours (skip) | **87.87** | 69.29 | **59.12** | **87.65** | **32.41** | **12.34** | **47.88** |

*Table 2.* Performance of Llama-3-8B-Instruct experiments on RULER tasks (64K context).

| Method | Avg. | S-1 | S-2 | S-3 | MK-1 | MK-2 | MK-3 | MV | MQ | VT | CWE | FWE | QA-1 | QA-2 |
|---|---|---|---|---|---|---|---|---|---|---|---|---|---|---|
| Standard | 57.3 | 76.8 | 99.4 | 67.2 | 84.2 | 31.0 | 0.0 | 85.2 | 87.1 | **70.4** | 7.9 | 54.7 | 46.8 | 33.6 |
| LongCE | 52.5 | 98.2 | 99.2 | 74.6 | 93.2 | 18.4 | 0.0 | 90.3 | 89.9 | 4.9 | 4.9 | 18.9 | **54.6** | 35.6 |
| PoSE | 67.5 | 99.6 | 100 | 75.6 | 90.2 | 70.6 | 6.6 | 86.2 | 90.5 | 61.4 | 22.7 | 79.8 | 53.8 | **40.2** |
| Ours (cyclic shift) | **71.3** | 100 | 100 | **81.6** | 86.8 | **88.4** | **27.4** | **93.1** | 93.6 | 67.1 | 16.1 | 80.7 | 51.6 | 41.0 |
| Ours (skip) | 69.3 | 100 | 100 | 75.0 | **94.0** | 82.6 | 20.4 | 88.6 | **94.5** | 43.9 | **26.1** | **82.5** | 53.2 | 40.0 |

index perturbations strengthens long-range evidence use.

We would like to note that we do not aim to claim that one perturbation is uniformly best across all tasks, as they represent different trade-offs. In practice, we recommend the skip variant because it is less destructive and better preserves order-sensitive behavior compared to cyclic (Section 3.6).

**Qwen-3-4B.** On Qwen, our method achieves the strongest RULER performance among single-technique baselines at both 128K and 256K (Table 3). Relative to Llama, the absolute gains in the RULER average are smaller, largely because several tasks are already near ceiling at these lengths, which compresses the unweighted average. The improvements are most visible on the harder retrieval variants (Appendix B.1).

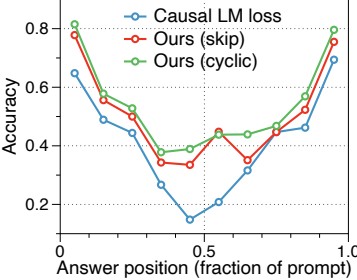

*Figure 3.* Positional sensitivity on RULER NIAH MultiKey-2 for Qwen-3-4B.

Figure 3 provides a finer-grained diagnostic by plotting accuracy as a function of normalized answer position on RULER NIAH MultiKey-2 for Qwen-3-4B. The standard baseline exhibits a pronounced U-shaped pattern, with a mid-context accuracy drop, consistent with the "lost-in-the-middle" phenomenon (Liu et al., 2024). Both variants of our method improve accuracy across positions and notably lift performance in the middle of the sequence.

In addition to RULER, we evaluate on HELMET at 128K (Table 3). Our single-technique variants are competitive and achieve the best HELMET average among the single-technique methods.

Finally, we consider a combined variant, *Ours (skip) + LongCE*, since the two techniques are largely orthogonal in motivation and potentially complementary. LongCE (Fang et al., 2025) reshapes the CLM learning signal by reweighting tokens to emphasize those benefit more from longer context, whereas our method regularizes the model to be invariant to RoPE-index perturbations. Combining them (LongCE reweighting on the CLM term plus our KL regularizer) aims to jointly improve the training signal and reduce positional sensitivity. Empirically, the combined variant achieves the strongest overall performance: it attains the best RULER score at 256K and the best HELMET average at 128K (Table 3).

### 3.3. Length Extrapolation via YaRN

Beyond in-distribution evaluation at the training window, we test whether positional-robustness regularization improves *length extrapolation*, i.e., performance when the input exceeds the training context length. We evaluate the checkpoints with YaRN (Peng et al., 2024) for extension at longer windows: 128K/256K for Llama and 512K/1M for Qwen.

As shown in Tables 4 and 5, standard fine-tuning degrades

*Table 3.* Average performance of Qwen-3-4B under different sequence lengths on RULER and HELMET (128K) benchmarks.

| Method | RULER | | HELMET | | | | |
|---|---|---|---|---|---|---|---|
| | 128K | 256K | RAG | ICL | LongQA | Rerank | Avg. |
| Standard | 74.45 | 67.01 | 51.54 | 86.08 | 56.02 | 13.57 | 51.80 |
| LongCE | 76.87 | 68.01 | 52.88 | 87.24 | 56.00 | **15.53** | 52.90 |
| PoSE | 75.47 | 65.48 | 52.04 | 87.16 | 55.47 | 12.51 | 51.79 |
| Ours (cyclic shift) | **78.37** | 67.91 | 53.00 | 86.52 | 54.48 | 13.58 | 51.89 |
| Ours (skip) | 77.51 | 68.10 | 53.96 | 87.40 | 57.23 | 13.39 | 52.99 |
| Ours (skip) + LongCE | 77.50 | **68.95** | **54.25** | **87.56** | **57.23** | 15.32 | **53.59** |

*Table 4.* Extrapolation performance of Llama using YaRN at extended context lengths.

| Method | 128K | 256K |
|---|---|---|
| Standard | 42.03 | 23.90 |
| LongCE | 46.54 | 26.91 |
| PoSE | 58.21 | 43.18 |
| Ours (cyclic shift) | **60.89** | **45.33** |
| Ours (skip) | 59.43 | 44.74 |

*Table 5.* Extrapolation performance of Qwen using YaRN at ultra-long context lengths.

| Method | 512K | 1M |
|---|---|---|
| Standard | 58.11 | 45.31 |
| LongCE | 60.52 | 47.53 |
| PoSE | 58.00 | 44.62 |
| Ours (cyclic shift) | 60.27 | 49.56 |
| Ours (skip) | **60.73** | **50.41** |
| Ours (skip) + LongCE | 60.13 | 50.02 |

*Table 6.* Average performance of Qwen-3-4B after SFT under different sequence lengths on RULER, HELMET, and LongBenchV2 (LBv2) benchmarks.

| Method | RULER | | HELMET | LBv2 |
|---|---|---|---|---|
| | 128K | 256K | | |
| Standard | 75.52 | 64.44 | 52.61 | 27.63 |
| LongCE | 76.95 | 65.92 | 53.36 | 31.73 |
| PoSE | 76.22 | 63.88 | 52.53 | 28.83 |
| Ours (skip) | 77.28 | 66.86 | 53.48 | 29.40 |
| + LongCE | **78.68** | **67.15** | **54.02** | **32.22** |

tion tuning, performance on instruction-style benchmarks is heavily confounded by formatting and instruction-following ability.

The improvements from RoPE-perturbed self-distillation largely persist or are even slightly enlarged after SFT: our method remains stronger than Standard and baselines on RULER and HELMET, and the combined variant (Ours + LongCE) achieves the best performance across benchmarks. Together, these results suggest that our training method improves underlying long-context capabilities rather than producing gains that are washed out by SFT.

### 3.5. Short-Context Performance

While our objective targets long-context robustness, it is important to ensure that it does not degrade short-context capability. Table 7 reports standard downstream benchmarks. Across both model families, our method preserves short-context performance and remains comparable to other baselines, suggesting that the gains at long context are not achieved by trading off general language understanding.

### 3.6. Ablations and Analysis

We next present a set of targeted ablations and controls designed to pinpoint which components drive the gains. Specifically, we (i) ablate key objective-design choices, (ii)

sharply under extrapolation, especially for Llama. In contrast, our method consistently achieves the best performance at extended lengths across both model families, indicating that encouraging invariance to RoPE index perturbations during training improves robustness when pushing beyond the trained window.

### 3.4. Does the Gain Survive Instruction SFT?

In practical pipelines, long-context continued pretraining is often followed by supervised fine-tuning (SFT) on instruction data. To test whether our gains persist after SFT, we perform one epoch of SFT on Tulu-v3 (Lambert et al., 2024) starting from the long-context-trained Qwen checkpoints, and evaluate at 128K/256K on RULER and HELMET. We additionally report scores on LongBench-v2 (Bai et al., 2024), a challenging long-context benchmark that covers more realistic long-context tasks (Table 6). We report LongBench-v2 only after SFT because, before instruc-

*Table 7.* Downstream task performance across models under different methods.

| Model | Method | MMLU | HellaSwag | Winogrande | OpenBookQA | Avg. |
|-------|--------|------|-----------|------------|------------|------|
| Llama | Base | 64.7 | 75.6 | 71.4 | 43.0 | 63.7 |
|       | Standard | 61.7 | 78.7 | 72.5 | 43.4 | 64.1 |
|       | LongCE | 62.9 | 78.7 | 72.2 | 43.8 | 64.4 |
|       | PoSE | 62.2 | 78.7 | 72.0 | 43.6 | 64.1 |
|       | Ours (cyclic shift) | 61.4 | 78.4 | 72.4 | 44.2 | 64.1 |
|       | Ours (skip) | 61.2 | 78.3 | 72.2 | 43.6 | 63.8 |
| Qwen  | Base | 68.4 | 68.4 | 65.8 | 40.6 | 60.8 |
|       | Standard | 68.6 | 70.6 | 67.8 | 40.4 | 61.8 |
|       | LongCE | 68.4 | 71.1 | 68.2 | 40.4 | 62.0 |
|       | PoSE | 68.6 | 71.1 | 68.8 | 40.6 | 62.3 |
|       | Ours (cyclic shift) | 68.2 | 70.3 | 68.0 | 40.6 | 61.8 |
|       | Ours (skip) | 68.5 | 71.2 | 68.9 | 39.4 | 62.0 |
|       | Ours (skip) + LongCE | 68.0 | 69.9 | 67.1 | 40.6 | 61.4 |

*Table 8.* Ablation of objective design and comparison with generic regularization on Llama-3-8B-Instruct (RULER, 64K).

| Method | Avg. |
|--------|------|
| Baseline | 47.9 |
| Ours | **59.2** |
| w/ forward KL | 57.8 |
| w/ CLM | 56.5 |
| w/o random skip | 55.1 |
| Dropout | 48.9 |
| Attention noise | 49.6 |

*Table 9.* Effect of skipping range $Y$ on RULER performance at 64K context (Llama-3-8B-Instruct).

| Skipping Range | RULER (64K) |
|----------------|-------------|
| 32K | 59.63 |
| 64K | 59.23 |
| 128K | 58.88 |

compare against two-pass baselines that add an extra consistency loss without perturbing RoPE, and (iii) study how performance varies with perturbation strength and with alternative RoPE-index transformations. Unless otherwise noted, all ablations are run on Llama-3-8B-Instruct at 64K and reported at an early checkpoint (200 steps) to surface qualitative trends while keeping compute affordable.

**Objective design.** Our objective makes two key choices: (i) using *reverse* KL with the standard view as a stop-gradient teacher, and (ii) sampling the skip parameters $(s, y)$ per sample to cover diverse positional variations. Table 8 shows that both choices are important. Replacing reverse KL with forward KL reduces performance, and replacing distillation with CLM training on the perturbed view (i.e., treating the perturbed view as additional training data) also underperforms. Finally, using a *fixed* skip (e.g., $s = y = 32K$) is worse than sampling $(s, y)$, suggesting that stochasticity improves coverage over positional shifts and robustness.

**RoPE perturbation vs. generic two-pass regularization.** A natural question is whether our improvements come simply from adding an extra forward pass with a consistency loss, rather than from perturbing positional indices. To test this, we compare against generic two-pass controls

that do *not* modify RoPE (Table 8): (i) dropout-based consistency that matches predictions across two stochastic passes, and (ii) Gaussian noise injected into attention scores. Both controls provide at most marginal gains over the baseline and remain far below our RoPE-perturbed objective. This indicates that the benefit is not explained by generic ensembling/noise-robustness effects; explicitly perturbing the positional representation is crucial.

**Perturbation strength is not overly sensitive.** We examine sensitivity to the magnitude of the skip perturbation by varying the skipping range while keeping all other settings fixed. As shown in Table 9, performance remains relatively stable across skip lengths, indicating that our method is not overly sensitive to the precise choice of skipping range. Notably, larger skips do not monotonically improve performance: excessively large perturbations can slightly degrade accuracy, suggesting that overly aggressive positional shifts may begin to distort useful positional structure.

**Alternative RoPE perturbations: structure-preserving shifts work best.** Our framework is agnostic to the specific RoPE-index transformation, but not all perturbations are equally effective. To probe what properties matter, we evaluate several alternative perturbation schemes under the *same* self-distillation objective (Table 10). Removing positional encoding entirely in the perturbed view (**NoPE**) is highly destructive and substantially degrades performance, indicating that the teacher signal alone cannot compensate for eliminating positional structure. More aggressive perturbations that

*Table 10.* Ablation of RoPE perturbation schemes on Llama (RULER, 64K).

| Method | Avg. |
|---|---|
| Baseline | 47.9 |
| Ours | **59.2** |
| NoPE | 29.5 |
| Random dilation | 53.4 |
| Chunked permutation | 46.2 |

disrupt global order, such as **chunked permutation** of large index blocks , provide little benefit and can hurt accuracy. In contrast, **random dilation** (randomly scaling RoPE indices by a factor in $[0.5, 2]$) yields moderate gains but still trails our skip-based shift. Overall, these results suggest that effective perturbations should introduce meaningful positional variation while preserving the local and monotonic structure of the underlying token order—a criterion naturally satisfied by skip-based shifts.

*Table 11.* Effect of the KL coefficient $\lambda$ on Llama-3-8B-Instruct (RULER, 64K).

| Method | RULER (64K) |
|---|---|
| Standard | 47.9 |
| Ours (skip, $\lambda = 0.2$) | 54.0 |
| Ours (skip, $\lambda = 0.5$) | 60.0 |
| Ours (skip, $\lambda = 1$) | 59.2 |
| Ours (skip, $\lambda = 2$) | 58.9 |

**Ablation on the KL coefficient $\lambda$.** We vary the KL coefficient $\lambda$ on Llama, and report RULER performance at 64K (Table 11). The main experiments apply $\lambda = 1$, where we do not extensively tune. The results show that the method is not overly sensitive to the precise choice of $\lambda$, with the best results obtained around $\lambda \in [0.5, 1]$. This trend is intuitive: when $\lambda$ is too small, the consistency signal is weak and the objective behaves closer to standard CLM training; when $\lambda$ is too large, optimization may overemphasize matching the perturbed and standard views at the expense of the main language-modeling objective.

**Order-sensitive NIAH.** A natural concern is whether encouraging invariance to RoPE index shifts could harm tasks that genuinely require order or positional information. To probe this trade-off, we introduce an order-sensitive synthetic evaluation at 64K context, NIAH-position. Unlike standard needle-in-a-haystack retrieval (a single key–value pair), we consider a multi-value variant where the same key appears multiple times in the haystack, each time paired with a different value. The query then requests an ordinally specified value (e.g., the first/second/third/last occurrence), explicitly testing whether the model can track relative ordering among repeated mentions.

*Table 12.* Accuracy on the 64K NIAH-position benchmark (multi-value ordinal retrieval) on Llama-3-8B-Instruct.

| Method | 64K NIAH-position |
|---|---|
| Baseline | 44.0 |
| LongCE | 56.2 |
| PoSE | 48.2 |
| Ours (cyclic shift) | 42.0 |
| Ours (skip) | **59.4** |

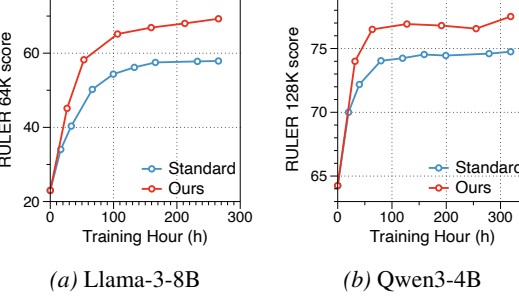

*(a)* Llama-3-8B      *(b)* Qwen3-4B

*Figure 4.* RULER performance versus wall-clock training time under matched compute.

Table 12 shows that our skip-based variant achieves the best performance, indicating that RoPE-perturbed self-distillation does not degrade—and can even improve—ordinal retrieval in this setting. A key reason is that the skip-based perturbation is *order-preserving*: it changes RoPE indices while keeping the token order unchanged, so the relative ordering of repeated key occurrences remains consistent. In contrast, the cyclic-shift variant underperforms, suggesting that perturbations that more strongly disturb cross-view positional alignment can conflict with strict ordinal queries. Overall, these results highlight that our framework can remain compatible with position-sensitive behaviors when using structure-preserving perturbations, while motivating careful perturbation design for order-dependent tasks.

### 3.7. Compute-Matched Training Efficiency

Our method adds one additional forward pass per step to compute the distillation loss. In practice this yields approximately a 1.6× wall-clock overhead per step compared to Standard CLM training. To determine whether the gains can be explained by increased compute, we compare against a compute-matched baseline: we train the Standard model for 1.6× more steps so that wall-clock time is matched, with other hyperparameters stay the same. Figure 4 plots RULER performance versus training time for both model families.

Across both Llama and Qwen, additional training provides limited benefit once Standard fine-tuning plateaus, whereas RoPE-perturbed self-distillation converges to a

higher-performing checkpoint under the same wall-clock budget. This indicates that the improvements stem from a stronger training signal induced by RoPE perturbations, not merely from spending more compute.

From a practical standpoint, this data efficiency is particularly important because high-quality long-context data is limited and expensive to obtain. By improving how such data is leveraged during training, our method achieves stronger long-context performance while incurring justified computational overhead.

## 4. Related Work

**Positional encoding.** Positional encoding design and RoPE extension strategies are a major line of work for pushing transformers beyond their pretraining context window. A prominent family of methods directly modifies the RoPE frequency schedule or scaling rule to reduce out-of-distribution (OOD) behavior at long positions, including NTK-/ABF-style scaling (Xiong et al., 2024), interpolation-based extensions such as YaRN (Peng et al., 2024), and follow-ups that refine the interpolation dynamics (Wang et al., 2024). Other approaches search for or parameterize frequency rescaling more flexibly, such as LongRoPE (Ding et al., 2024), or multi-scale positional encoding designs (Zhang et al., 2024). Complementary to rescaling, several works explore alternative positional encoding choices and hybrids, including NoPE (Kazemnejad et al., 2023) and hybrid attention/position strategies that interpolate between RoPE and NoPE (Yang et al., 2025b). While these approaches focus on changing the positional encoding function to better fit the pretrained regime, our method keeps the underlying RoPE mechanism intact and improves long-context behavior by regularizing robustness to controlled RoPE-index perturbations during adaptation, making it largely orthogonal and potentially composable with the above extensions.

**Objective-level training and position simulation for long context.** Another line of work improves long-context generalization by modifying the training objective or by simulating long-range positional relationships during training. On the objective side, LongCE (Fang et al., 2025) reshapes the learning signal by reweighting tokens in the CLM loss; its motivation—emphasizing tokens that benefit from long context—is largely orthogonal to our focus on positional robustness. On the simulation side, several approaches (Zhu et al., 2023; Ruoss et al., 2023; Wu et al., 2024a;b; Hu et al., 2025) manipulate or sample positional indices so the model encounters long-range relative offsets even when trained on shorter windows, effectively treating position transformation as a form of perturbation-based training. In contrast to methods tied to a single positional simulation mechanism, we propose a RoPE-perturbed self-distillation framework that is agnostic to the specific perturbation rule: we instanti-

ate it with skipping and also show that other perturbations, such as shift-based schemes, can be effective. This shifts the objective from simulating long offsets to learning invariance to RoPE-index perturbations, which is the property we directly target. Empirically, we directly compare against LongCE and PoSE, and consistently achieve better long-context performance.

**Inference-time mitigation of positional bias.** A separate line of work targets positional weakness at inference time rather than during training. One approach scales a position-sensitive hidden-state channel to reduce position bias at test time (Yu et al., 2025). Another remaps or shifts RoPE indices during inference so that poorly trained positions are replaced by better-conditioned ones (An et al., 2024). These methods are complementary as they intervene only at test time, whereas our method changes the training objective and uses standard indexing at evaluation time.

## 5. Conclusion

Standard long-context fine-tuning can remain positionally brittle: accuracy still strongly depends on where the relevant evidence appears, which is undesirable for tasks where evidence placement varies widely. We propose RoPE-Perturbed Self-Distillation, a regularizer that creates alternative "views" of the same sequence by perturbing RoPE indices (e.g., skip-based or cyclic shifts) and enforces prediction consistency via a KL-based self-distillation loss. Across Llama-3-8B and Qwen-3-4B, this improves long-context benchmarks—especially harder retrieval/composition tasks—while preserving short-context performance, and the gains persist after instruction SFT. Moreover, the method strengthens length extrapolation under YaRN-based extension, suggesting the learned invariance transfers beyond the trained window; ablations indicate the key is using structure-preserving perturbations and a targeted consistency signal rather than generic two-pass regularization.

While we instantiate our approach with RoPE because it is widely used in modern open-source LLMs, the underlying principle is more general. Analogous perturbations could be defined for other positional encoding mechanisms, making the framework not inherently RoPE-specific. The type of perturbation can also be further adapted for specific settings. Exploring such non-RoPE instantiations, as well as task-aware perturbation distributions for structured long-context pipelines, is a promising direction for future work.

## Impact Statement

This paper presents work whose goal is to advance the field of Machine Learning. There are many potential societal consequences of our work, none which we feel must be specifically highlighted here.

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

# A. Experimental Details

## A.1. Training protocol and compute

We follow the ProLong training recipe (Gao et al., 2025) and train each method for 1000 steps under fixed-length sequences at the target window.

- **Llama-3-8B-Instruct (64K).** Global batch size 4M tokens/step on 16×A100 GPUs, totaling 4B tokens.

- **Qwen-3-4B (256K).** Global batch size 8M tokens/step on 32×A100 GPUs, totaling 8B tokens.

## A.2. Baselines and implementation notes

**Standard.** CLM-only long-context continued pretraining under the same data and compute budget.

**LongCE.** We reimplement LongCE (Fang et al., 2025). Following the default configuration, we use a 4K short-context reference length and compute weights using a sliding window of 1K. All other training settings match the Standard baseline.

**PoSE.** PoSE (Zhu et al., 2023) can be interpreted as training CLM on a perturbed RoPE index assignment. We implement PoSE by applying the same perturbation magnitude as our skip-based method and optimizing CLM on the perturbed view (i.e., without the KL consistency term), keeping all other settings identical.

# B. Extended Results

## B.1. Qwen RULER task breakdowns at 256K

Table 13 reports the task-level breakdown for Qwen-3-4B at 256K context, including the combined variant **Ours (skip) + LongCE**. Relative to Standard, our method yields its clearest improvements on more challenging retrieval/composition subsets, while LongCE tends to help complementary subsets; combining them provides the strongest average.

*Table 13.* Performance of Qwen-3-4B on RULER tasks at 256K context.

| Method | Avg. | S-1 | S-2 | S-3 | MK-1 | MK-2 | MK-3 | MV | MQ | VT | CWE | FWE | QA-1 | QA-2 |
|---|---|---|---|---|---|---|---|---|---|---|---|---|---|---|
| Standard | 67.01 | 100 | 98.8 | 97.2 | 85.8 | 41.2 | 25.6 | 78.15 | 89.4 | 95.4 | 8.4 | 78.2 | 41.0 | 32.0 |
| LongCE | 68.01 | 100 | 97.8 | 99.0 | 84.2 | 45.4 | 17.2 | 90.1 | 89.3 | 90.8 | 8.4 | 78.2 | 47.4 | 36.4 |
| PoSE | 65.48 | 100 | 99.8 | 97.6 | 84.8 | 46.8 | 23.6 | 73.3 | 86.7 | 86.8 | 1.7 | 77.6 | 41.2 | 31.4 |
| Ours (cyclic shift) | 67.92 | 100 | 99.4 | 97.8 | 86.6 | 54.2 | 58.0 | 75.2 | 87.3 | 78.6 | 1.2 | 71.3 | 39.2 | 34.2 |
| Ours (skip) | 68.10 | 100 | 100 | 98.4 | 82.6 | 50.2 | 39.6 | 75.15 | 88.5 | 93.1 | 2.5 | 77.6 | 43.6 | 34.0 |
| Ours (skip) + LongCE | **68.95** | 100 | 97.2 | 99.4 | 83.2 | 45.8 | 28.4 | 96.5 | 90.0 | 85.8 | 4.6 | 83.1 | 46.4 | 36.0 |

## B.2. Mix-length Training

*Table 14.* Average performance of mix-length-trained Llama-3-8B-Instruct under different sequence lengths on RULER and HELMET (64K) benchmarks.

| Method | RULER | | HELMET |
|---|---|---|---|
| | 32K | 64K | Avg. |
| Standard | 79.94 | 52.48 | 46.73 |
| Ours (skip) | **85.27** | **65.28** | **48.80** |

**Mix-length training: gains persist under realistic length mixtures.** While our main results focus on fixed-length training for controlled comparison, practical long-context continued pretraining often uses a mixture of sequence lengths to better cover diverse inputs. We therefore train Llama-3-8B-Instruct under a ProLong-style mix-length schedule up to 64K and evaluate on RULER (32K/64K) and HELMET (64K). Table 14 shows that our skip-based self-distillation remains consistently better than Standard fine-tuning across evaluation lengths. This indicates that the benefit of positional-robustness regularization is not specific to fixed-length training and carries over to more realistic training regimes.

## B.3. Evaluation Variance Across Random Seeds

For RULER, each subtask contains 500 examples, giving 6.5K examples across the 13 subtasks. For HELMET, we follow the official evaluation configuration. We rerun evaluation for the Llama-3-8B-Instruct checkpoints with 3 random seeds and report the mean and standard deviation on the main aggregate metrics.

*Table 15.* Evaluation variance across random seeds on Llama-3-8B-Instruct. We report mean performance with standard deviation in parentheses.

| Method | RULER 32K | RULER 64K | HELMET |
|---|---|---|---|
| Standard | 85.26 (0.08) | 57.32 (0.12) | 44.26 (0.17) |
| Ours (cyclic shift) | 87.29 (0.08) | 71.25 (0.14) | 46.17 (0.09) |
| Ours (skip) | 87.82 (0.09) | 69.32 (0.09) | 47.81 (0.13) |

# C. Extended Related work discussion

**Long Context Training Recipe.** Many works improve long-context capability by continued pretraining and careful "recipe + data engineering," rather than modifying model architecture. Recent studies (Gao et al., 2025; Xiong et al., 2024; Fu et al., 2024) emphasize that strong performance at 32K–128K depends on training protocol choices and long-sequence data construction/mixtures, showing that simply increasing the maximum length is often insufficient without targeted data and training design. Complementary to recipe tuning, curriculum-style approaches such as GrowLength (Jin et al., 2023) progressively increase sequence length to improve training efficiency and stability when scaling context. Pushing beyond 128K, Xu et al. (2025) demonstrates that ultra-long training can be made practical with system-aware and data-aware design choices. In our experiments, we follow a ProLong-style training recipe to ensure a strong and fair baseline, and we focus on an objective-level modification that is plug-compatible with these continued-pretraining pipelines.

**Other directions for long-context capability.** Beyond mentioned directions, a broad set of methods improve long-context performance by changing the architecture or the workflow used to process long inputs. On the architecture side, many works target the quadratic cost of attention via sparse / structured attention patterns (Beltagy et al., 2020; Zaheer et al., 2020), training-free KV retention/streaming strategies (Xiao et al., 2023; Han et al., 2024; Zhang et al., 2023), and recurrent structure (Bulatov et al., 2023). On the workflow side, long-context ability can be improved by using external memory and retrieval mechanisms (Wu et al., 2022), retrieval-augmented generation (RAG) pipelines (Jiang et al., 2024; Asai et al., 2024). These directions primarily address efficiency or pipeline design, whereas our method targets positional robustness via an objective-level regularizer, and is largely complementary to architectural and workflow-based advances.

