# OpenReview forum: "Shuffle the Context: RoPE-Perturbed Self-Distillation for Long-Context Adaptation"
_ICML.cc/2026/Conference — ICML 2026 regular_

### Official Review · Reviewer_9Udz · 2026-02-21

**Soundness:** 3
**Presentation:** 3
**Significance:** 2
**Originality:** 2
**Overall Recommendation:** 4
**Confidence:** 2

**Summary:**

This paper addresses the sensitivity of large language models to Rotary Position Encoding (RoPE) in long-context tasks by proposing a self-distillation regularization method that constructs perturbed views of RoPE indices. By enforcing prediction consistency between a standard view and a perturbed view, the method enhances positional robustness. Experiments on Llama-3 and Qwen-3 show improvements on long-context benchmarks like RULER and HELMET. Crucially, this is achieved without degrading performance on short-context tasks such as MMLU.

**Compliance With Llm Reviewing Policy:**

Affirmed.

**Final Justification:**

The authors' response has further reinforced my previous judgment, so I would like to maintain the positive rating of "weak accept."

**Key Questions For Authors:**

1. Is this method applicable to other positional encoding methods ?
2. Can the performance be validated on more challenging long-context mathematical or code generation tasks (e.g., MATH with long reasoning chains or long-context code completion), as this would strengthen the evidence for real-world applicability.

**Limitations:**

yes

**Strengths And Weaknesses:**

**Strengths:**

1.  The approach of constructing perturbed views of RoPE indices for self-distillation to enhance model robustness to positional encoding is novel and theoretically valuable.
2.  The method is comprehensively evaluated on multiple models (Llama, Qwen) and long-context benchmarks (RULER, HELMET), with detailed analytical experiments, lending high credibility to the conclusions.
3.  The method significantly improves long-context performance without sacrificing short-context capabilities and is compatible with existing training pipelines, demonstrating strong practical potential.

**Weaknesses:**

1.  Each training step requires an additional forward pass (for the perturbed view), increasing computational time and leading to higher costs in large-scale training.
2.  While two perturbation methods (skip-based and cyclic shifts) are proposed, there is a lack of theoretical guidance on which is most effective, potentially requiring tuning for different models.
3.  The paper does not deeply explore whether the method might introduce potential risks or performance fluctuations in tasks that are extremely sensitive to strict sequential order, such as code generation or mathematical reasoning.

---

> ### Author Rebuttal · Authors · 2026-03-30
>
> Thank you for the positive review and insightful comments! Please find our response to your concerns below:
>
> > **W1. Each training step requires an additional forward pass (for the perturbed view), increasing computational time and leading to higher costs in large-scale training.**
>
> Thanks for pointing this out, we agree that the extra forward pass increases computational cost. We discussed the cost and also include a compute-matched analysis to test whether this overhead is worthwhile in Section 3.7. In practice, the method adds about 1.6x wall-clock time per step, so we compare against a Standard baseline trained for 1.6x more steps with the same total wall-clock budget. The result is that additional CLM training alone yields only limited gains after plateau, while our method consistently reaches a better final checkpoint on both model families. Thus, the added computation is justified by improved training efficiency rather than simply spending more compute.
>
>
> > **W2. While two perturbation methods (skip-based and cyclic shifts) are proposed, there is a lack of theoretical guidance on which is most effective, potentially requiring tuning for different models.**
>
> Thank you for this comment. We would like to clarify that the goal of this paper is not to identify a universally optimal perturbation, but to show that RoPE perturbation itself is a useful and generic principle for long-context adaptation under the self-distillation framework. The two variants we study (skip-based and cyclic shift) are intended to demonstrate this generality rather than to have a single fixed design.
>
> We believe that theoretical guidance for selecting the best perturbation is difficult for modern LLMs due to the complexity of optimization. Therefore, we provide empirical guidance through ablations (Section 3.6 line 358): effective perturbations should introduce positional variation while preserving local order and structure, whereas overly destructive perturbations are less effective. This is also supported by Appendix B.3 (line 670), where the skip-based perturbation performs best on an order-sensitive benchmark because it preserves token order. Therefore, we view skip-based as the practical default choice, while cyclic shift serves as a stronger alternative stress test within the same framework.
>
>
>
>
> > **W3/Q2. The paper does not deeply explore whether the method might introduce potential risks or performance fluctuations in tasks that are extremely sensitive to strict sequential order, such as code generation or mathematical reasoning. Can the performance be validated on more challenging long-context mathematical or code generation tasks (e.g., MATH with long reasoning chains or long-context code completion), as this would strengthen the evidence for real-world applicability.**
>
> To address this concern, we additionally evaluate the post-SFT long context adapted Qwen-3-4B model on the long-reasoning AIME 26 benchmark. The results are:
> | Method        | AIME 26 Avg@16 | AIME 26 Pass@16 |
> | - | - | - |
> | standard | 22.50 | 46.67|
> | Ours (skip)   | 24.17| 50.00|
>
> These results suggest that our method does not hurt performance on long-reasoning math task, and in fact provides a small improvement. We would also like to clarify that the skip-based perturbation preserves the original token order and only modifies the RoPE indices, which makes it naturally more compatible with tasks that depend on strict sequential structure. By contrast, we agree that the cyclic-shift variant may be less suitable for such settings. This is consistent with our order-sensitive evaluation in Appendix B.3, where cyclic shift degrades performance, while the skip-based variant remains the strongest.
>
>
>
> > **Q1. Is this method applicable to other positional encoding methods?**
>
> Conceptually, the method is not specific to RoPE. Our core idea is to create alternative positional “views” of the same sequence and enforce prediction consistency across them. As long as a positional encoding scheme allows meaningful positional perturbations while keeping the token content, the same regularization principle can be applied. We focus on RoPE in this paper because it is the most widely used positional encoding in modern LLMs and supports such perturbations in a particularly natural and controlled way; extending the framework to other positional encodings is an interesting direction for future work.

---

> > ### Author Rebuttal · Reviewer_9Udz · 2026-04-01
> >
> > I'd like to thank the authors for responding my questions. I'd like to maintain my positive rating.

---

### Official Review · Reviewer_H3pU · 2026-03-05

**Soundness:** 4
**Presentation:** 4
**Significance:** 4
**Originality:** 3
**Overall Recommendation:** 5
**Confidence:** 4

**Summary:**

The paper addresses a key drawback of large language models adapted to long contexts: positional instability, which implies that even after fine-tuning on long sequences, the model's accuracy can vary significantly depending on the location of the relevant information in the input. This is problematic for applications such as RAG or multi-document reasoning, where evidence can appear at any position. The authors propose a new training method to make models more robust to the absolute position of information, encouraging them to rely on semantic cues rather than unstable positional cues. To achieve this, the authors introduce an additional LLM forward pass with the same input but augmented with RoPe positional embeddings and add a KL distance component between the logits of the augmented and unaugmented forward passes. The resulting model is more robust to positionality and RoPe scaling.

**Compliance With Llm Reviewing Policy:**

Affirmed.

**Key Questions For Authors:**

1. It would be useful if the authors provided an ablation for the lambda parameter for the KL loss component. Intuition suggests that its choice can influence the final performance of the method.

**Limitations:**

yes

**Strengths And Weaknesses:**

**Strengths:**
- The authors address a pressing issue affecting the performance of LLM on long-context networks.

- The proposed method is quite original and easy to implement.

- The authors conduct a detailed analysis of two types of RoPe augmentations and arrive at a well-founded conclusion about which one is optimal.

- The proposed method allows for a more efficient scaling of RoPe to long-context networks using YaRN by a factor of 2-4.

- The authors demonstrate that their method is compatible with another method, LongCE, and that their combination yields improved results.

- The paper presents convincing results from measurements on various long-context benchmarks demonstrating the effectiveness of the method.

- The authors demonstrate that the effect of their training method persists even after SFT on Tulu-3.

**Weaknesses:**
- No evidence is provided regarding whether the method's effect persists after the RLHF stage (PPO/DPO/GRPO).

- It would be useful to understand how the method performs on Mixture-of-Experts if the authors could provide a similar ablation.

---

> ### Author Rebuttal · Authors · 2026-03-30
>
> Thank you for the positive review and helpful comments! Please find our response to your concerns below:
>
> > **W1. No evidence is provided regarding whether the method's effect persists after the RLHF stage (PPO/DPO/GRPO).**
>
> We agree that evaluating whether the gains persist after a downstream RLHF stage would be valuable. In the current submission, we examine robustness to an additional SFT stage (Section 3.4), and the gains remain after this extra stage. This suggests that our method improves underlying long-context capability rather than yielding only transient training-time effects. A full PPO/DPO/GRPO study is somewhat orthogonal to the main focus of this paper, since RLHF introduces an additional task- and reward-dependent objective on top of long-context adaptation, and whether gains persist can depend substantially on the specific RL setup, including the data distribution and reward design. That said, recent evidence [1] suggests that on-policy RL may preserve prior capabilities better than SFT by favoring smaller-KL updates, so existing literature does not indicate that our gains should be especially fragile under RLHF. We will clarify this scope in the revision and leave a full RLHF evaluation as important future work.
>
> > **W2. It would be useful to understand how the method performs on Mixture-of-Experts if the authors could provide a similar ablation.**
>
> We include preliminary results on a Mixture-of-Experts model, gpt-oss-20B, below. We follow the same training data and hyperparameter setup as in the Qwen-3-4B experiments, and extend the model to 256K context. Due to resource constraints, these results are still preliminary, so we report an early checkpoint at 200 training steps (1B token). We observe an improvement on the RULER 256K benchmark, suggesting that the benefit of our method is not limited to dense models.
> | Method        | RULER (256K) |
> | - | - |
> | Standard      | 64.63 |
> | Ours (skip)   | 67.03 |
>
> > **Q1. It would be useful if the authors provided an ablation for the lambda parameter for the KL loss component. Intuition suggests that its choice can influence the final performance of the method.**
>
> We thank the reviewer for this suggestion. To address this, we add an ablation over the coefficient $\lambda$ on Llama-3-8B under the same early-checkpoint setting as Table 9. The results are shown below:
> | Method        | RULER (64K) |
> | - | - |
> | standard | 47.90 |
> | Ours (skip, lambda=0.2)   | 54.02|
> | Ours (skip, lambda=0.5)   | 60.01|
> | Ours (skip, lambda=1)   | 59.23|
> | Ours (skip, lambda=2)   | 58.99|
>
> These results show that the method is not overly sensitive to the choice of $\lambda$. The best results are obtained around $\lambda \in [0.5, 1]$. This trend is also intuitive: when $\lambda$ is too small, the consistency signal is weak and the method behaves closer to standard CLM training; when $\lambda$ is too large, the model may overemphasize matching the perturbed and standard views at the expense of the main CLM objective.
>
> **Reference**
>
> [1] RL's Razor: Why Online Reinforcement Learning Forgets Less

---

> > ### Author Rebuttal · Reviewer_H3pU · 2026-04-01
> >
> > I would like to thank the authors for answering my questions and providing supporting ablation experimental results, therefore I maintain my positive rating.

---

### Official Review · Reviewer_4JoH · 2026-03-08

**Soundness:** 3
**Presentation:** 3
**Significance:** 3
**Originality:** 3
**Overall Recommendation:** 5
**Confidence:** 3

**Summary:**

This paper focuses on the problem of long-context scenarios in large language models. The authors show that LLMs are sensitive to the position of key information in long contexts, whereas this position can be fragile in RAGs, coding libraries, etc. To solve this, the authors introduce RoPE-perturbed self-distillation, which builds two views from a single input: the original view serves as a teacher, while the other is generated by perturbing the RoPE index. The author adds a reverse KL loss to the original causal LM loss, encouraging the model to be consistent across different length dependencies within the same context. Experiments show that models with short context lengths can achieve outstanding long-context abilities without sacrificing short-context abilities.

**Compliance With Llm Reviewing Policy:**

Affirmed.

**Final Justification:**

I appreciate the authors’ detailed rebuttal and clarifications. My earlier concerns have been adequately addressed, especially regarding benchmark coverage beyond NIAH-style settings and the explanation of the method's mechanism. These responses further strengthen my confidence in the paper.

Overall, I view this as a simple, effective, and practically relevant contribution to long-context adaptation, and I maintain my positive rating in support of acceptance.

**Key Questions For Authors:**

1. Is this method trained with full parameters, or did the author use LoRA?

**Limitations:**

See Weaknesses

**Strengths And Weaknesses:**

## Strengths

1. This paper has a clear structure and is easy to follow.

2. The method is simple and effective. Meanwhile, the authors also provide ablation studies to prove its effectiveness. The idea of self-distillation and perturbation is interesting and has the potential to boarden to more areas.

3. I appreciate the experiment of using NIAH to show the position sensitivity in long-context scenarios, which gives a clear background for demonstration.

## Weaknesses

1. The benchmarks used in experiments are basiclly NIAH-related tasks, since the authors state the problem exist extremely clearly in industry-level RAG and repo understanding, some experimennts with that kind of dataset seems necessary.

2. I suggest the authors provide mechanism explanations of why this method works.

---

> ### Author Rebuttal · Authors · 2026-03-30
>
> Thank you for the positive review and comments! Please find our response to your concerns below:
>
> > **W1 The benchmarks used in experiments are basically NIAH-related tasks, since the authors state the problem exist extremely clearly in industry-level RAG and repo understanding, some experimennts with that kind of dataset seems necessary.**
>
> We agree that NIAH-style evaluation alone would be insufficient for a method motivated by industry long-context use cases. Therefore, our benchmarks are not limited to NIAH: beyond RULER, we also evaluate on HELMET, which includes practical long-context categories such as RAG and ICL, and on LongBench-v2 after SFT, which covers more realistic tasks including code/repository QA. These results were intended to test whether the gains transfer beyond synthetic retrieval settings, and we will revise the paper to make this broader evaluation coverage more explicit.
>
>
> > **W2. I suggest the authors provide mechanism explanations of why this method works.**
>
> We hypothesize that the method works by regularizing the model against brittle dependence on the RoPE index assignment. RoPE shifts mainly perturb the more position-sensitive, high-frequency components of the RoPE transformation while preserving token content and much of the coarse sequence structure. Enforcing consistency between the standard and perturbed views therefore encourages the model to rely more on semantic evidence and coarse positional structure, rather than overfitting to absolute-position cues.
>
> We also provide several empirical results that support this interpretation. As shown in Figure 3, our method substantially reduces positional sensitivity on NIAH-style retrieval, with especially clear improvements in the middle of the context, where the standard model exhibits a pronounced lost-in-the-middle pattern. In Figure 2, our attention analysis further shows that the model assigns more attention mass to longer distances and exhibits a more uniform long-range attention distribution. We will revise the paper to make this mechanism and supporting evidence more explicit.
>
>
>
> > **Q1. Is this method trained with full parameters, or did the author use LoRA?**
>
> We train the model with full parameters. We will include this detail in the experiment setting section.

---

> > ### Author Rebuttal · Reviewer_4JoH · 2026-04-01
> >
> > Thank you for your detailed rebuttal! I have no more questions and think this is an outstanding work and should be accepted.

---

### Official Review · Reviewer_nAbm · 2026-03-14

**Soundness:** 3
**Presentation:** 3
**Significance:** 3
**Originality:** 3
**Overall Recommendation:** 3
**Confidence:** 4

**Summary:**

The paper proposes RoPE-Perturbed Self-Distillation, a training regulariser for improving long-context LLMs. The key observation is that standard long-context fine-tuning leaves models brittle to where relevant evidence appears in the context. The fix is simple: during training, create an alternative "view" of each sequence by shifting its RoPE positional indices, then penalise the model (via KL divergence) for producing different predictions under the two views, encouraging it to rely on semantic content rather than absolute position. Experiments on Llama-3-8B and Qwen-3-4B show consistent but modest gains on long-context benchmarks (RULER, HELMET) and improved length extrapolation, without hurting short-context performance.

**Compliance With Llm Reviewing Policy:**

Affirmed.

**Final Justification:**

I stand by my original  assessment.

**Key Questions For Authors:**

- The gains are genuinely modest in several places. On Qwen at 256K the improvement is only 1.09 points on RULER. On HELMET the picture is quite inconsistent, the skip variant sometimes trails even Standard on individual categories. In Table 13 the cyclic-shift variant scores 42.0, below the 44.0 baseline, on ordinal retrieval. I am not sure how big the test sets are; I suspect they are fairly large, but it would be good to include error bars. The comparison between skip and cyclic variants is also hard to interpret without variance estimates,  in Table 1 cyclic beats skip on RULER-64K (71.30 vs 69.29) but skip wins on HELMET, and without error bars you can't tell if either difference is reliable.

-  I did not quite understand why the Ours (skip) + LongCE is reported *only* for Qwen and not for Llama.

- Evaluating only on RULER and HELMET (synthetic benchmarks) limits the practical story; LongBench-v2 is included only post-SFT and the gains there are modest.
- Can you please discuss how your paper conceptually and practically differs from:
https://arxiv.org/pdf/2406.02536v3 and
https://arxiv.org/pdf/2410.18745

**Limitations:**

There is no section specifically dedicated to limitations. The final paragraph briefly acknowledges that the method is RoPE-specific and suggests extending it to other positional encoding mechanisms as future work.

**Strengths And Weaknesses:**

- The paper addresses a real issue for LLMs expected to perform tasks over long contexts demonstrating the positional brittleness issue in Figures 1a and 3.

- The core idea requires no architectural changes,  just a modified training objective with one extra forward pass. It's  compatible with existing long-context training recipes like ProLong, which makes it practically appealing. The comparison against generic two-pass regularisation (dropout, attention noise) in Table 8 rules out the explanation that any consistency regulariser would work, and isolates the RoPE perturbation itself as the key ingredient.

- The specific combination of RoPE index perturbation with self-distillation (KL consistency) is new. Prior work perturbed positions but treated them as additional training data under CLM; using a teacher-student consistency loss across perturbed and unperturbed views is a nice idea.

- Figure 4 directly addresses the concern that gains come simply from extra compute, showing the method reaches a higher performance ceiling under matched wall-clock time.

- The improvements survive instruction SFT (Table 6), hold across two quite different model families, and transfer to length extrapolation beyond the training window, all of which suggest the method is improving something fundamental rather than overfitting to benchmark format.

- The paper is well-written and easy to follow.

- PoSE (Zhu et al., 2023) already does positional skip-wise training;  the main difference as far as I can tell is the KL distillation term rather than just CLM on the perturbed view.

- Randomised positional encodings for length generalisation (Ruoss et al., 2023)  covers much of the same intuition about exposing models to diverse positional offsets during training.  The paper feels more like a well-executed incremental contribution than a fundamentally new idea

---

> ### Author Rebuttal · Authors · 2026-03-30
>
> Thank you for your insightful comments. Please find our response below:
>
> > **W1/W2: [1] already does positional skip-wise training. [2] covers much of the same intuition. The paper feels more like a well-executed incremental contribution.**
>
> We agree that our work is related to prior position-perturbation methods [1, 2]. But we believe the contribution is more than incremental. As you noted in your review, *“Prior work perturbed positions but treated them as additional training data under CLM; using a teacher-student consistency loss across perturbed and unperturbed views is a nice idea.”* This is the key distinction of our method. The importance of this change is that it shifts the learning objective: prior work uses perturbed indices to expose the model to longer positional offsets for better length generalization, whereas our objective directly trains the model to be robust to RoPE perturbations, this improve **both** in-distribution performance and extrapolation. This change is also supported by experiments. Our method consistently performs better than PoSE [1] on both models. PoSE even hurt the performance a bit on Qwen-3-4B.
>
> > **Q1a. The gains are genuinely modest in several places. On HELMET the picture is quite inconsistent.**
>
> While the gains are smaller in some settings, we believe the main results are meaningful rather than modest overall. On Llama, our method improves RULER 64K by over 12%, improves RULER 32K and HELMET average by 2.6%. On Qwen, it improves RULER 128K by over 3%. On RULER 256K, the improvement is smaller, but it is still a meaningful gain over a benchmark with 13 tasks.
>
> For HELMET, we do not expect every subtask to improve uniformly; it is a broad multi-skill benchmark where per-subtask trade-offs are common, so the key metric is the overall average. On both models, our method achieves the best average. On Qwen, the skip variant improves RAG, ICL, and LongQA, and is only 0.18 below Standard on Rerank. We would also emphasize that the gains persist after SFT.
>
> > **Q1b. In Table 13, the cyclic-shift variant underperforms baseline.**
>
> Table 13 is included as a diagnostic to test whether RoPE perturbations can hurt order-sensitive tasks. The key takeaway is that this failure mode appears for the more disruptive cyclic-shift variant, but not for the structure-preserving skip variant.
>
> > **Q1c. How big are the test sets? Need to include error bars. The comparison between skip and cyclic variants is hard to interpret.**
>
> For RULER, each subtask has 500 examples (6.5K total), so the average is reasonably stable; for HELMET, we follow the original configuration. We do not claim one variant is uniformly better across all tasks, only that they represent different trade-offs and prove the generality of the framework; in practice we recommend the skip variant because it is less destructive. We also reran evaluation on Llama with multiple seeds and observed small stds:
>
> |Method|RULER 32K|RULER 64K|HELMET|
> |-|-|-|-|
> |Standard|85.26 (0.08)|57.32 (0.12)|44.26 (0.17)|
> |Ours (cyclic)|87.29 (0.08)|71.25 (0.14)|46.17 (0.09)|
> |Ours (skip)|87.82 (0.09)|69.32 (0.09)|47.81(0.13)|
>
>
> > **Q2. Why the Ours (skip) + LongCE is reported only for Qwen and not for Llama.**
>
> On Llama, LongCE itself is weaker than the Standard baseline, so we do not combine it with our method.
>
> > **Q3. Evaluating on RULER and HELMET limits the practical story; LongBench-v2 is included only post-SFT and the gains there are modest.**
>
> We chose RULER and HELMET because they are broad long-context benchmarks, and HELMET in particular targets practical settings such as RAG and ICL rather than being a trivial toy evaluation. We report LongBench-v2 only after SFT because pre-SFT scores on instruction-style benchmarks are heavily confounded by instruction-following and formatting ability rather than isolated long-context capability. We also would not characterize the LongBench-v2 gain as modest: +1.8 is meaningful on this benchmark. For example, on the longbench v2 leaderboard, the gap between Qwen 3 235B A22B and Qwen 3 32B is 1.6, despite the large model size difference.
>
> > **Q4. Can you please discuss how your paper conceptually and practically differs from [3] and [4]**
>
> We clarify the distinction below and will include it in the revision.
> Both [3] and [4] are inference-time methods for positional weakness: [3] scales a positional hidden-state channel, and [4] shifts RoPE indices at inference. In contrast, our method is a training-time objective that learns robustness by matching standard and RoPE-perturbed views. Practically, their methods require test-time intervention, while ours uses the standard model at inference.
>
>
> [1] PoSE: Efficient Context Window Extension of LLMs via Positional Skip-wise Training
>
> [2] Randomized Positional Encodings Boost Length Generalization of Transformers
>
> [3] Mitigate Position Bias in LLMs via Scaling a Single Hidden States Channel
>
> [4] WHY DOES THE EFFECTIVE CONTEXT LENGTH OF LLMS FALL SHORT?

---

> > ### Author Rebuttal · Reviewer_nAbm · 2026-03-31
> >
> > I will adjust my score.

---

### Decision · Program_Chairs · 2026-04-30

**Decision:**

Accept (regular)

**Comment:**

This paper studies positional brittleness in large language models for long-context tasks, proposing a RoPE-Perturbed Self-Distillation training regularizer to encourage reliance on semantic content rather than absolute position. The reviewers find that the paper addresses a highly relevant problem using a simple method without any architectural changes, and successfully improves long-context abilities without sacrificing short-context performance. They also raise concerns on the modesty of some performance gains, the computational overhead of the extra forward pass, and the need for broader evaluation on non-synthetic or strictly order-sensitive tasks.

During rebuttal, the authors provided multiple-seed variance estimates, new empirical data on the AIME benchmark, preliminary results on a Mixture-of-Experts architecture, and a compute-matched analysis to justify the training overhead. The reviewer generally acknowledge that the concerns have been fully addressed. The AC also agrees that the paper makes a solid and practically impactful contribution to the community. Therefore, an acceptance is recommended.